# A Systems Thinking Approach towards Single-Use Plastics Reduction in Food Delivery Business in Thailand

**Boonchanit Wongprapinkul [1] and Sujitra Vassanadumrongdee [2,*]**

1  Environment, Development and Sustainability Program, Graduate School, Chulalongkorn University, Bangkok 10330, Thailand; boonchanit.w@gmail.com
2  Environmental Research Institute, Chulalongkorn University, Bangkok 10330, Thailand
*  Correspondence: sujitra.v@chula.ac.th

**Abstract:** Sustainable consumption and production (SCP) is largely influenced by dynamics in the market system. Hence, this study adopts the systems thinking approach as a tool to understand the non-linearity and complexity of sustainable practices. The food delivery business has accelerated the plastic waste problem, especially during COVID-19 where restaurant dine-in was limited. This study aims to identify high leverage points, which contribute to system intervention strategies to improve Thailand's single-use plastics (SUP) waste situation caused by the food delivery sector. Semi-structured interviews were conducted with relevant stakeholders ($n = 14$). A qualitative system dynamics model analysis and thematic analysis suggest that the key leverage points include (1) benefit alignment among all stakeholders, (2) cost minimization and profit maximization, (3) laws and regulations, (4) postconsumption waste management systems, and (5) research and development. In addition, to stimulate policy initiatives, this study suggests that system intervention strategies should include behavioral instruments (setting default and eco-labelling), market-based instruments (green-packaging procurement and subsidies), and system and infrastructure provision (waste management systems and deposit return models). Partnerships, research and development, and laws and regulations are proposed as supporting measures.

**Keywords:** systems thinking; system dynamics; leverage points; food delivery; sustainable practices; single-use plastic; COVID-19

## 1. Introduction

From a business perspective, the food delivery business is considered as an on-demand fulfilment player in the e-commerce ecosystem. It is a multidimensional platform business that needs network inputs from several stakeholders. At the early stage, the food delivery companies use a 'Loss Leader' strategy where they accept an initial financial loss to create lively market environments and to implant new consumption behaviors. To this day, the platform continues to expand its business operations to best meet ever-increasing consumer needs. The coronavirus disease 2019 (COVID-19) event has delayed most economic activities. However, the food delivery business thrived in response to emerging needs and rapid digital disruption. The changing market environment has forced businesses to design consumer-centric solutions on a day-to-day basis to keep up with the demand. The legal factor that accelerated the growth of food delivery services was the government measures that prohibited restaurant dine-in; only takeaway and delivery orders were allowed. The World Bank Group [1] pointed out that COVID-19 has significantly increased the consumption of food delivery. The Standard Wealth [2] reported a substantial change in online food-ordering habits in a global landscape. In Tokyo, the number of people who ordered food via a mobile application doubled in 2020. Taiwan, Hongkong, Japan, China, Singapore, Indonesia, Malaysia, the Philippines, and India have witnessed the post-COVID-19 growth of the food delivery business and are all pushing investment and innovation into it. In

Thailand, food delivery businesses exhibited their financial success during the past few years since launching, and consumers promptly responded to this new lifestyle solution. Influenced by COVID-19, this business experienced a 150% growth rate during the first half of 2020. The order reached 66–68 million transactions with a 78–84% growth rate in 2020 [3]. Tanakasempipat [4] added that COVID-19 has multiplied the transaction amount by 50–400%. Due to rapid expansion, several new businesses entered the market during 2020–2021. Most of them were introduced as subsidiaries under existing big brands in industries such as e-commerce, banking, telecommunications, and airlines.

Despite the impressive financial figures, this convenience-based business poses a threat to the planet's carrying capacity. It exacerbates the lazy economy and throw-away culture, which intensifies several environmental impacts. One of them is single-use plastic (SUP) waste. The COVID-19 outbreak has escalated the plastic waste problem globally. Thailand's Pollution Control Department, in 2021 [5], estimated that, as people were asked to stay home, food delivery consumption increased by 30%, thus generating a 15% increase in overall plastic waste [6]. With the assumption that each food delivery order generates 4–11 pieces of plastic, it is estimated that 250–560 million pieces of SUP are generated each year from this business [5,7–10]. The top three plastic waste from food delivery services are plastic bags, hot-and-cold food bags, and plastic food containers [11]. Moreover, COVID-19 has delayed the effort in campaigning against SUP in many countries. Massachusetts, New Hampshire, San Francisco, and many other states suspended a ban on polystyrene foam containers, as well as imposed a temporary ban on reusable shopping bags and allowed retailers to give out free plastic bags. Scotland postponed its packaging deposit return scheme (DRS) to late 2022. India suspended the ban on SUP bags and bottles. The United Kingdom removed SUP bag charges from delivery services [12]. Thailand, according to the roadmap, should have been phasing out SUPs by 2020. Nevertheless, plastic bans were relaxed during the pandemic. The bring-your-own campaign was paused, and most restaurants rejected private containers.

This study views SUP problems from the system perspective. Because sustainable development is about balancing three interconnected pillars, a non-linear approach can enhance the understanding of multidimensional problems in a dynamic world. Systems thinking, or organic thinking, has gained more popularity in assisting holistic policy decisions. It aims to leverage the system towards a self-organizing stage so that it can continue operating in a more sustainable direction. Theoretically, the systems of provision theory develops from the narrowness of the mainstream neo-classical utilitarian approach. The systems theory explains that every day (un)sustainable consumption practices are supported or constrained by the existing systems [13]. It believes that a meso-level examination of the system needs to replace a sole focus on micro- or macro-level agencies because a single dimension analysis cannot depict the interrelations between structure and agencies [14–16]. Berkhout [17] also pointed out that unless the system is designed to reinforce green consumption, it is not likely that we can expect a green lifestyle. During the COVID-19 pandemic, Thai consumers were obligated to buy takeaway meals. Choice of packaging was also limited. Therefore, this research relies on systems thinking and systems dynamics, which is a popular approach under systems thinking. It contributes to the holistic perspective of SUPs reduction, considering variables at production, consumption, and postconsumption stages from both policy and businesses outlooks.

The research process is shown in Figure 1. Targeted sustainable initiatives were reviewed through the interviews. Under the systems thinking approach, data from the interviews went through thematic analysis, which produced system themes. In parallel, a qualitative system dynamics model (QSDM) was drafted based on the researchers' initial review of current situations, comprising three causal loop diagrams (CLDs). Through semi-structured interviews, stakeholders were asked to review and iterate each CLD, which was then integrated into the QSDM. The interviews were conducted again for the final QSDM iteration. These analysis results contributed to the system's leverage points, which were interpreted into system intervention strategies. This paper is structured as follows.

Section 2 introduces systems thinking and system dynamics theory, together with the tools and approaches adopted in this research. Section 3 reviews initiatives in the food sector, focusing on plastic waste from the food delivery service. It explores market governance initiatives and private-led initiatives. Four initiatives were selected as a subject of study. Section 4 provides the results from the thematic analysis and QSDM. Themes and diagrams are obtained in this section. Section 5 discusses leverage points and system intervention strategies. Section 6 draws conclusions and research contributions.

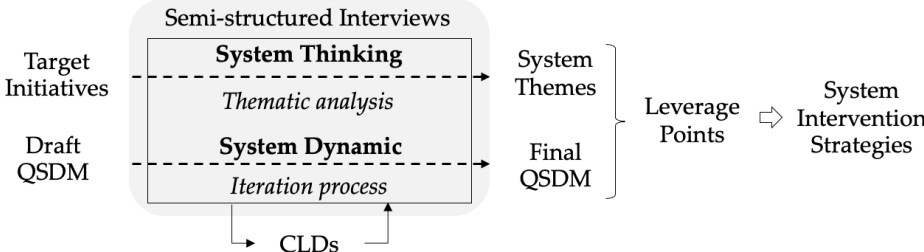

**Figure 1.** Research process flowchart.

## 2. Systems Thinking and System Dynamics: Theory and Approach

### 2.1. Systems Thinking Theory

Systems thinking is an approach to exploring factors, actors, and their multilayered relationships. A complex issue needs to be tackled on a systems-thinking basis rather than on a compartmentalized-thinking basis due to the non-static characteristics of the systems. While the elements may be easy to identify, the interconnections within the subsystems are less obvious and need to be studied at a profound level [18]. For these reasons, systems thinking considers subtle factors, so-called, 'mental models' that may not be explicitly presented but can influence the system, such as consumer perception, incentives, and recognition [19,20]. Apart from mental models, the systems thinking framework includes 'systemic structures', 'patterns', and 'events' [21,22]. The systemic structure explains relationships within the systems, system structure, and the rules that govern them. Patterns refer to system behavior and practices. Events explicitly involve the reality of practices at the tip of the iceberg where most managerial decisions happen. Therefore, the results from systems thinking analysis can contribute to the enhanced understanding of the target issue and the linkages among them, thus allowing us to identify measures and strategies to improve system behaviors [20,23]. Systems thinking has successfully assisted the context-specific business strategies and development policies formulation [24]. Systems thinking is found to be applicable within sustainability studies such as green business practices, business models for sustainability, customer expectation management, waste and water management, recycling and reuse behavior, the penetration of electric vehicles, food waste, and household energy consumption, e.g., [25–29]. From the sustainable consumption outlook, the system operation depends on how each actor views value. The unaligned values may produce undesired system behavior [18]. Today's market competitively offers more choices at a more affordable cost for greater convenience. Thus, it is interesting to explore sustainable consumption from a systems thinking perspective.

Under the systems thinking approach, thematic analysis was adopted as a data analysis method for data obtained from semi-structured stakeholder interviews. It was defined by [30] as a method to identify, analyze, and report patterns (themes) within the data set. Thematic analysis looks beyond the surface meanings into the true implications of each assigned code. Codes are a list of items from the data that have a recurring pattern relating to the issues within the research area. After individual data is gathered and thoroughly interpreted, codes are developed during the data-coding process to identify key ideas and issues that emerge within the data set. Codes can be drawn directly from the verbal language, or from the more conceptual language. Occasionally, new codes can emerge, and old codes can be reviewed as the coding process is conducted [31]. The process continues

until the data coding is saturated. Codes are then combined and categorized into themes in which codes are analyzed in a more systematic way [32].

*2.2. System Dynamics and Leverage Points Identification*

Systems thinking can be applied through the development of an ecosystem or system modelling such as system dynamics modelling. A system dynamics (SD) technique was developed by [20] under a systems thinking concept that aimed to understand the dynamics of the system. This technique was later adopted as a methodology to assist many development issues including sustainability transformation. In this research, SD is carried out through the qualitative system dynamics model (QSDM). The need to use qualitative system dynamics as a research methodology was originally stressed by [33] in 1985. While the numerical equation aims for analytical traceability, systems thinking aims for a more qualitative and illustrative understanding of trajectories and results from model simulations [24,34]. In addition, Coyle and Exelby [35] studied different types of SD models and pointed out that qualitative analysis led to a deep understanding of the system, which could assist in effective policy decisions. As claimed by [24], a good model incorporates all factors deemed to be important despite the availability of its numerical data to estimate the parameters. In this study, the model was constructed via Vensim software (Vensim® PLE 9.0.0x64 (x64), Build: TK-210730.00048, Ventana Systems, Inc., Bangkok, Thailand), which is a common SD simulation modelling tool.

The concept of a system's leverage point was initially discussed by [36] as a place in the system that, if adjusted, can lead to significant improvements. As part of systems theory, leverage points can be found through a systems thinking approach using the system dynamics model as a tool. There is no shortcut to finding high leverage points in the system, as the complex system is counterintuitive. Researchers must understand the insights of system arrangement through model iteration with stakeholders. Past research found that the leverage points to sustainable development involve interventions at mental levels, such as concerns and norms; systemic structure levels, such as market mechanisms; and event levels, such as behavioral mechanisms [21,37–39].

In particular, SD was used as a tool to analyze the plastic waste problem to create policy mix in India [40], waste management and extended producer responsibility (EPR) strategy in Indonesia [41,42], environmental impact in China [43], and technological solutions [44]. However, as most studies focus on policy-level intervention, only one study used a quantitative approach. Focusing on the postconsumption stage, these papers found that laws and regulations such as charging disposal fees and imposing bans, recycling systems and subsidies, research and development (R&D), industry standardization, and EPR were the variables that represent leverage points in the system.

## 3. Single-Use Plastic Reduction Initiatives in the Food Sector

*3.1. Market Governance Initiatives*

Viewing the market as a system with failures, a sustainable market is hindered by conflicts of interest among the actors and between the actors and structure. First, 'measurement failure' refers to the fact that economic prosperity is a development priority in every country. Second, 'self-regulation failure' refers to the voluntary nature of sustainability practices among private corporations. Therefore, governance is a key component that regulates the system to produce the desired outcome. This section reviews relevant laws, regulations, policies, and practices. The European Commission has introduced the directive on Single-Use Plastic Products 2019/904, which has been transposed into national law. Measures such as market bans, consumption reductions, and EPR were proposed. Under the Packaging Waste Directive, EPR requires producers to cover the costs of waste prevention, waste collection and treatment, litter clean-up, data gathering, and consumer awareness-raising. Likewise, the Chinese government has introduced measures that promote alternative packaging in the food delivery and e-commerce sectors. It also calls for optimized business models, and waste segregation and recycling facilities [45]. Under the

World Wide Fund for Nature (WWF)'s PACT (Plastic ACTion) initiative, food delivery platforms, Grab, Foodpanda, and Deliveroo in Singapore signed the pledge and implement initiatives such as setting 'no cutlery' as a default, providing restaurant guidelines, exploring alternatives, and pilot tests with returnable packaging. The extended producer responsibility (EPR) concept receives significant attention. Although EPR originally focused on the producers' responsibility of their product beyond the production stages, its definition has been extended to cover responsibilities at other stages. In this study, EPR covers the responsibility of food delivery companies, food retailers and restaurants, which facilitate intermediate supply and demand for packaging.

In Thailand, as food delivery business is relatively new to the market, the governances are not yet comprehensive. In alignment with the national strategy, the Plastic Waste Management Road Map, 2018–2030, sets out to ban plastic bags less than 36 microns in thickness, styrofoam food containers, SUP cups, and straws by 2022. However, the ban is not legally binding. The tax incentives for bio-packaging also received little interest. As COVID-19 underlines the urgency of the problems, the Pollution Control Department (PCD) under the Ministry of Natural Resources and Environment and another 14 agencies, including major food delivery platforms, signed a Memorandum of Understanding (MOU) in August 2020 to tackle the SUP problem in food delivery. The main strategies include 1. making the 'no cutlery' function the default option in the mobile application, 2. introducing the restaurants' eco-label in the mobile application, 3. providing an incentive (discount) or disincentive (charges) to consumers to choose greener packaging, 4. promoting environmentally friendly restaurants, 5. disseminating information among supply chain actors, particularly consumers, 6. providing incentives to restaurants for a greener packaging choice, and 7. providing subsidies for green packaging. Additionally, Deutsche Gesellschaft für Internationale Zusammenarbeit (GIZ), in partnership with PCD, initiated the 'rethinking plastics—circular economy solutions to marine litter' [45]. The project promotes 'reducing single-use plastics in food delivery and takeaway' which suggests 1. tax and non-tax incentives, 2. packaging standards (e.g., materials, recyclability, and biodegradability), 3. take-back systems, and 4. waste management systems improvement. Although these voluntary approaches have contributed to aggregated awareness, their contribution to SUP reduction efforts is still limited.

### 3.2. Private-Led Initiatives

As SUP waste problems from food delivery services have received more public attention, private sustainability projects were increasingly initiated as part of the corporate responsibility program. Globally, the most common SUP reduction strategy is to set 'no cutlery' as a default option as a nudge to consumer behavior. Some platforms, such as Foodpanda in Singapore and Hong Kong, and Meituan Waimai in China, offer rewards for no-cutlery orders via a point system. Occasionally, a fee is applicable for additional bags or containers requested. Deliveroo, Foodpanda, and Meituan Waimai in the UK, Australia, Singapore, and China establishes partnerships with food packaging companies to offer discounts for eco-friendly packaging to their merchant partners. Some of the platforms bring reusable packaging into their business model through DRS. Meituan Waimai, a major food delivery platform in China, works with the China Packaging Federation on the provision of alternative packaging. Foodpanda and Deliveroo in Singapore partnered with bearPack, a packaging company, to develop a reuse system. In addition, the Alternative Materials Tool system was developed to assist the restaurant partners in choosing packaging materials with the least environmental impact. Likewise, other service providers are looking for sustainable solutions through business partnerships and government support in terms of eco innovation.

Moreover, there are new players whose business models are developed on a sustainable basis. These companies are either established in the form of the delivery platform itself or in the supporting services. Go Box partners with more than 100 local food vendors in Portland and San Francisco Bay, United States and distributes returnable food containers

and coffee cups for takeaway orders. A membership system was established together with more than 30 drop sites. Go Box also encourages office buildings to have their own drop sites. DeliverZero in New York City, as well as Recircle in Switzerland established a reuse system for takeaway restaurants. Once the Tupperware is worn out, Recircle purchases them back from their restaurant partners at the same price and forwards them to the recycling system. Other food delivery startups that adopt deposit return models include DabbaDrop and Dabbawala, Deliveround, Sharepack, Vanilla Bean, Ozarka, Ozzi, reBOX, Yumiie, and Returnr, located in the UK, India, Belgium, Netherlands, Germany, US, Switzerland, and Australia. 90% of them establish deposit and rewards systems [46]. In India, Zomato and UberEats, responded positively to the government's SUP ban. However, the sustainable practice is hindered by the lack of alternative containers. In South Korea, a public-private agreement was formed to increase the recyclability of food packaging and promote reusable containers in restaurants [47].

In Thailand, food delivery platforms' actions towards plastic reduction include setting 'no cutlery' as a default, developing consumers feedback system, promoting green merchants, and procuring green packaging through partnership with 15–25% discount. Several startup platforms initiated the DRS business model, but the impact is still limited. Some local restaurants adopt the DRS within their neighboring ecosystem. 'Send plastic back home' is a project that is a partnership between the public, private sector, the social enterprise, and social groups. It conducts an information campaign on household waste separation and developed a plastic take-back system. Moreover, Central, a retail mall group, initiated the 'Rethink' project that accepts used (clean) SUP food containers in exchange for discounts in various stores.

### 3.3. Target Initiatives

The target initiatives were obtained from the review of concepts and cases of sustainability programs in food delivery businesses in Thailand and other countries. They were also derived from actual policies under the established guidelines that the Thai government and relevant stakeholders are currently working on implementing. This study analyzes four theory-based initiatives from a systems perspective with the aim of improving their penetration strategies considering the existing systems structure.

#### 3.3.1. No Cutlery Defaults

Platforms set 'no cutlery' as a default option for every transaction throughout the platforms. Consumers need to make a request if they want SUP cutlery. Theoretically, setting a default can be regarded as a nudge, which is a behavioral instrument in behavioral economics that could reduce the number of steps taken by consumers to opt for a greener choice.

#### 3.3.2. Packaging Procurement

Platforms establish partnerships with packaging suppliers and offer discounts to merchant partners. This initiative aims to change the system of practice from the supply side through the provision of affordable and greener packaging options. It holds the principle of corporate responsibility that the platforms are expected to absorb additional costs incurred from changing to green packaging to correct the pricing failure in the market.

#### 3.3.3. Labelling Program

Platforms add an 'eco-label' to restaurants that use environmentally friendly packaging. Additionally, the food delivery platforms can promote green restaurants as a category. Ideally, information provision can lead to better consumer decisions. From a marketing perspective, eco-labelling can act as a differentiation point for a restaurant [13]. From the policy viewpoint, informative instruments are endorsed by many development schools as one of the policy mechanisms that the governing actors could adopt to induce behavioral change.

### 3.3.4. Deposit Return Scheme (DRS)

Platforms develop a deposit return system as part of the EPR scheme. Consumers are required to roughly rinse their containers and return at the drop sites. Alternatively, they can make a pick-up appointment via mobile application. The platforms will then take back the containers to properly clean and reallocate back to the restaurants. Additionally, the government could provide support on systems and infrastructure. However, this scheme is challenging because the key 'reason to buy' of food delivery consumers is the 'convenience' attribute that the platforms offer. Besides, the overall economic return might be negative due to the high operation and logistics cost.

## 4. Results

Thematic analysis and QSDM were conducted through semi-structured interviews. The stakeholders included four parties ($n$ = 14); policy-level stakeholders ($n$ = 4), food delivery platforms ($n$ = 3), restaurant partners ($n$ = 5), and civil society organizations ($n$ = 2). Expert stakeholders were selected to enhance validity in the research process. The interview questions included topics about their practices and perspectives on SUP waste and waste management, what they expected to see from whom, and what could be improved (see Supplementary Materials).

### 4.1. Thematic Analysis

The semi-structured stakeholder interviews were conducted to gain an empirical and comprehensive understanding of how the plastic waste problems in the food delivery system can be improved. In addition, target initiatives were reviewed and evaluated under the situational context. Through thematic analysis, the data were transcribed and coded according to the keywords and issues that showed a pattern (see Supplementary Materials). A text analytics tool was used to capture patterns in the theme grouping process. As shown in Table 1, codes were then processed into themes, which were categorized into four sections according to the systems thinking framework. The sections included mental models, systemic structures, patterns, and events. Frequency of responses in codes under each theme are presented as 'score'. Similarly, Table 2 shows the themes of each target initiative. These results were analyzed in the leverage points identification process.

The mental models' themes revealed the insights of stakeholders, which represent the insights of the system, the reasons and motives behind sustainable and unsustainable practices. Notably, most themes were clustered at the systemic structure, which explains how the system could support or limit sustainable practices. Patterns involve themes that directly affect the rate of stock variable, which is the amount of SUP, as well as green packaging. Events pertain to the observable, current practices produced by the systems. Furthermore, it was found that stakeholders' conversations cluster in themes 2, 3, 5, 7, 9, 10, 15, 17 and 18. These themes were the areas that every stakeholder agreed upon.

Considering the situational context, Table 2 shows that the no-cutlery default faces practical limitations but is easy to implement and can save restaurants' costs. Packaging procurement is challenging due to the lack of practical packaging alternatives. Moreover, government subsidies are expected. The labelling program needs to be coupled with initiatives. Lastly, the DRS cannot be sustained by private entities alone. The system is complex and takes time to implement.

Apart from the qualitative reviews of target initiatives, stakeholders were asked to quantitatively evaluate each initiative on three criteria: expected outcome, resources required, and chance of success, on a ten-Likert scale. The average scores are presented in Table 3. The results revealed that the initiatives that received high-expected outcome, low resource required, and a high chance of success were the no cutlery default and eco-labelling initiatives.

**Table 1.** Thematic analysis results of the system.

| | System Theme | Explanation | Score |
|---|---|---|---|
| **Section 1: Mental Models** | | | |
| Theme 1 | Demand-driven, bottom-up approach towards sustainability | Promote awareness and behavioral change. Demand should be created before supply | 9 |
| Theme 2 | Incentives and cost-minimization principle | Incentives should be provided. Every stakeholder works on the cost-minimization principle | 14 |
| Theme 3 | WTP, awareness, and value gap | WTP, awareness, and value gaps are presented at every level, from suppliers to consumers | 11 |
| Theme 4 | Voluntary schemes do not work | Voluntary schemes have no significant result. Business fears losing its competitiveness. Mandatory waste management responsibilities should be applied at every level. | 6 |
| **Section 2: Systemic Structures** | | | |
| Theme 5 | Incentive alignment | Any sustainable initiative will be successful if every party is satisfied with the benefits received | 12 |
| Theme 6 | Internalization through incentives and disincentives | Internalization concepts should be considered through the provision of incentives and disincentives (tax and non-tax) | 9 |
| Theme 7 | Incentives for for-profit actors | Being profit-led, incentives are needed. Business cannot go on with projects that are not profitable | 10 |
| Theme 8 | Expectation on business responsibility | The business sector is expected to have a certain level of responsibility. However, under multi-stakeholder conditions, business alone cannot deliver significant change | 7 |
| Theme 9 | Standards, frameworks, regulations | The system needs standards, frameworks, and regulations for every stakeholder to ensure mutual direction | 11 |
| Theme 10 | Government subsidies | The government should provide subsidies as a pricing mechanism to push down the price and lift the demand for green packaging | 11 |
| **Section 3: Patterns** | | | |
| Theme 11 | Platforms and restaurants' role and relationships | Platforms and restaurants' roles, relationships, and the scopes of responsibility need to be clearly addressed | 7 |
| Theme 12 | Research and development (R&D) | R&D at the production stage should be promoted (e.g., mono-material packaging and alternative packaging) | 9 |
| Theme 13 | The current system is linear, not circular | The system is linear, not circular. The existing waste management system is not supportive of a circular economy | 7 |
| **Section 4: Events** | | | |
| Theme 14 | Customer-centric | Businesses need to focus on communication/consumer experience, impression, and satisfaction | 9 |
| Theme 15 | Lack of efficient postconsumption waste management system | Need an efficient postconsumption waste management system | 13 |
| Theme 16 | Niches as system disruptors | Sustainable niches such as startups and civil society organizations can disrupt the system with technologies and organization flexibility | 5 |
| Theme 17 | Have willingness but not equal capacity | The business sector has the willingness to act responsibly, but they (especially the restaurants) do not have equal capacity | 12 |
| Theme 18 | Lack of alternatives | Lack of alternatives (not practical, no economies of scale) | 12 |

**Table 2.** Thematic analysis results of the target initiatives.

| | Initiative Theme | Explanation | Score |
|---|---|---|---|
| **Initiative 1: No Cutlery Defaults** | | | |
| Theme 1 | Clear communication to consumers | To avoid the risk of complaints, opt-in and opt-out should be clearly communicated | 9 |
| Theme 2 | Practical limitation | Restaurants usually give out plastic cutlery regardless of consumers' requests | 12 |
| Theme 3 | Easy and cost-saving | Easy to do. Restaurants can save costs | 10 |

**Table 2.** *Cont.*

| Initiative Theme | | Explanation | Score |
|---|---|---|---|
| Initiative 2: Packaging Procurement | | | |
| Theme 1 | Consumer's packaging choice should not be restricted | Choices of packaging should be available, especially when consumer charges are involved | 8 |
| Theme 2 | Low availability of alternatives | Lack of greener alternatives for most types of meals | 11 |
| Theme 3 | Needs government subsidies | Small retailers cannot bear green packaging costs. Food delivery platforms cannot provide long-term subsidies | 10 |
| Initiative 3: Labelling Program | | | |
| Theme 1 | Easy to do but require green demand and supply | Without adjusting demand and supply, labelling may not lead to long-term changes | 9 |
| Theme 2 | Incentives to both restaurants and consumers | Incentives can draw restaurants and consumers to participate in the program | 12 |
| Initiative 4: Deposit Return Scheme (DRS) | | | |
| Theme 1 | High cost, not economically viable | This scheme requires high investment, expected economic return is low | 14 |
| Theme 2 | An inconvenient scheme in convenience-based business | While food delivery service offers convenience and speed, this scheme is inconvenient for every party | 13 |
| Theme 3 | Consumers' hygienic concerns | In the event of a pandemic, the reuse model may prove difficult | 9 |

**Table 3.** Target initiative evaluations.

| | Evaluation Criteria | | |
|---|---|---|---|
| | Expected Outcome | Resources Required * | Chance of Success |
| Target initiative | | | |
| 1. No cutlery default | 6.91 | 3.37 | 7.21 |
| 2. Packaging procurement | 5.49 | 8.08 | 5.34 |
| 3. Labelling program | 7.16 | 3.85 | 7.20 |
| 4. Deposit return scheme | 6.14 | 8.87 | 4.03 |

* Resource required was reversely interpreted.

### 4.2. Qualitative System Dynamics Model (QSDM)

To acquire the QSDM, first, the problems were identified. Second, the actors and their roles in the system were comprehended. Next, the behavior and trends of the system were observed. Then, causal loop diagrams (CLDs) were constructed, reviewed, and iterated. Lastly, CLDs were integrated into the comprehensive QSDM.

#### 4.2.1. Problem Identification

The first step of constructing QSDM is to identify problems in the system [26,38]. In this study, the problems that prevent the food delivery system from achieving sustainability can be identified as, first, consumers do not have enough information to make optimal decisions. Second, the price in the market does not reflect the true cost of the product, making the price of green packaging higher than its plastic alternatives. Moreover, the SUP waste problem is beyond the responsibility of a private company. Most of the sustainability programs are project-based and fail to deliver long-term results. The current systems and infrastructure are unsupportive of sustainable consumption and production. These limitations hinder sustainable consumption practices. The problem of SUP in the food delivery sector is context-specific and complex, so systematic analysis is needed to assist policy decisions. The leverage points identification process considered points that could alleviate these problems.

#### 4.2.2. Actors and Their Role in the System

The actors in this research include, first, the plastic producers, who design and innovate plastic packaging choices. Second, the food retailers and restaurants, who create

the intermediate demand for packaging. Third, the food delivery platforms, who could facilitate the supply and demand of SUPs. Fourth, the end consumers, who are faced with limited choice and are rarely able to choose packaging types. Lastly, the government, who could intervene in the system by offering incentives to influence the market into the preferred direction. In addition, the projects and initiatives developed from cooperation among the many actors, including sustainable niches, also play a role in shaping the market system. Each actor possesses different abilities, capabilities, and willingness to support greener consumption. The government sector performs a governance role by regulating the production, distribution, use, and disposal of plastic packaging. The business sector can also play a governance role through self-regulation by setting up a standard for the industry to improve the sustainable performance of the market. Business can improve the market environment in which competition among the corporates is value-driven, not profit-driven. The third sector refers to non-profit actors or civil society organizations that support sustainable initiatives. Additionally, factors involved in the system were derived from the analysis of stocks and flows, which are SUPs used in food delivery services and their journey from cradle to grave. Additionally, four target initiatives were included in the model as factors that are expected to influence the system.

### 4.2.3. Behavior-Over-Time Analysis

There is no official report on plastic waste generated from the food delivery service in particular. Therefore, secondary data were used to explain how variables behave over time. Several studies explore the potential long-term change in food consumption patterns influenced by COVID-19. The Boston Consulting Group [48] confirmed the long-term behavioral shift of US consumers away from restaurant dining, which was influenced by COVID-19. In Thailand, the usage of food delivery services dropped after the government eased the lockdown measures, but remained higher than before the pandemic [49,50]. Liu et al. [11] found a significant shift from eating out to online food delivery services among Thai consumers during COVID-19. Similarly, FirstCraft [51] found peak conversation about food delivery services among Thai consumers during the lockdown. These findings point to rising trends in food delivery services and the use of SUPs, with COVID-19 as a system catalyst. The food delivery systems' behaviors were driven by the changing market structure, which were caused by digital disruption and the government lockdown measures.

### 4.2.4. Causal Loop Diagram

A causal loop diagram (CLD) is the main tool in SD analysis. The CLD depicts the causal relationship between different elements in the system, which cannot be best explained in a simple linear manner, but rather in a more comprehensive loop. The causal loop diagram consists of arrows and signs, representing linkages and directions. Reinforcing loops (R) occur when change in one variable causes change in other variables in the same direction, thus reinforcing the loop. Balancing loops (B) occur when change in one variable causes change in other variables in the opposite direction, thus creating the balance of the loop. In this study, the stock variables are SUP packaging available in the market, SUP packaging stock at the restaurants, SUP stock with consumers, and SUP waste unmanaged in the environment.

To develop QSDM, each CLD was reviewed and iterated during the semi-structured interview until it gained acceptance among stakeholders. Three CLDs were presented according to the journey of stock variables, which were the amount of SUPs in the system. The first diagram shows causal linkages at the SUP production stage. The second and last diagrams represent the use stage, and the postconsumption stage. Each diagram underwent an iteration process by expert stakeholders. The final CLDs are shown in Figures 2–4 as follows: stock variables were framed, '#' was assigned to flow variables. Four target initiatives were shaded.

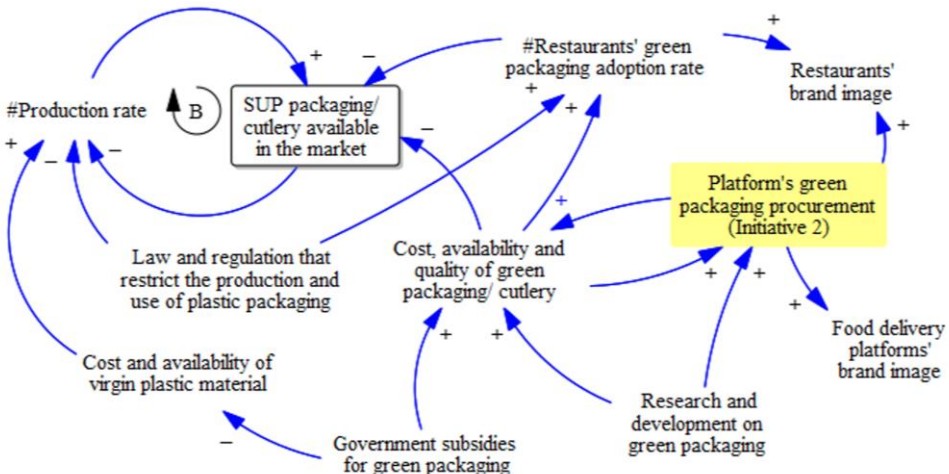

**Figure 2.** Causal loop diagram 1: production stage.

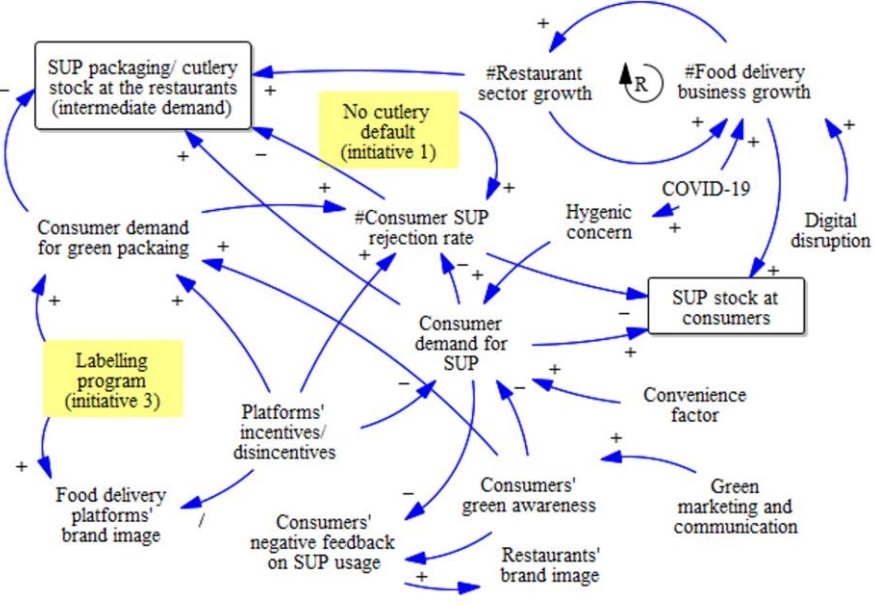

**Figure 3.** Causal loop diagram 2: use stage.

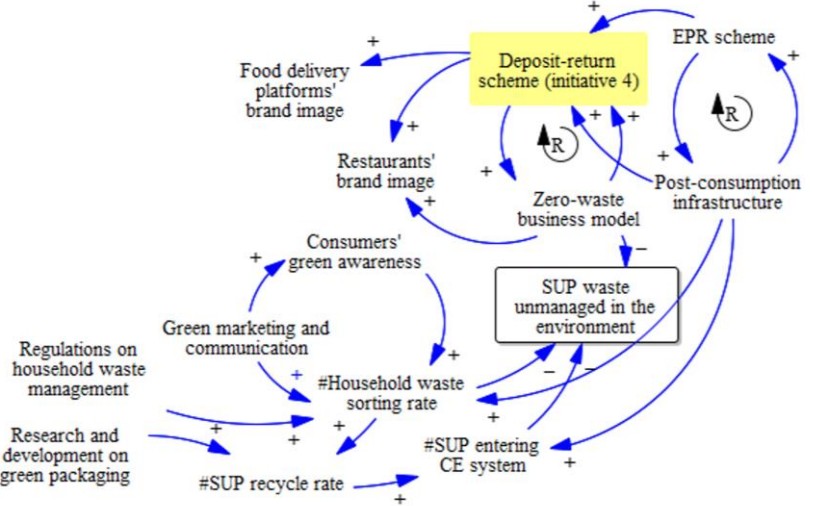

**Figure 4.** Causal loop diagram 3: postconsumption stage.

1. Production Stage

At the production stage (Figure 2), one balancing feedback loop can be identified. The availability of SUP packaging in the market, which is the stock variable, depends on the production rate. Likewise, if SUP packaging is oversupplied, the production rate needs to be slowed down. The production rate is also controlled by the cost and availability of virgin plastic, and by laws and regulations that restrict the production and use of conventional plastic packaging. However, as restaurants use more green packaging, the stock of SUP packaging in the market shrinks. The green packaging adoption rate is, in turn, reinforced by factors such as legal constraints of SUP, as well as the cost, availability, and quality of green packaging. Government subsidies, R&D, and packaging procurement initiatives directly improve the cost, availability, and quality of green packaging. Brand image can be improved through participation in a green packaging program. At this stage, it is observable that the cost and availability of green packaging is a key driver of restaurants' green packaging adoption, which directly influences market supply and production.

2. Use Stage

At the use stage (Figure 3), one reinforcing loop can be identified. Food delivery businesses undeniably accelerate the growth of the restaurant sector as it offers new sales channels, especially amid COVID-19. Likewise, as the restaurant sector grows, demand for food delivery services increases. Restaurants' SUP packaging stock depends on how much they can sell (growth), consumer packaging demand, and how often they reject unnecessary SUPs. Similarly, SUP stock at final consumers is determined by how frequently they order food delivery, how much they demand SUPs, and how often they reject SUPs. It is observable that environmental awareness, perceived brand image, convenience, and hygienic concerns are all underlying factors that influence the system at a mental model level through consumer demand, which directly influence stock variables. Furthermore, behavioral instruments such as incentives and nudges (default setting), as well as informative instruments such as labeling and communication can influence consumer demand and their behavior. However, while incentives reinforce brand image, disincentives may deteriorate brand image among some consumer groups.

3. Postconsumption Stage

At the postconsumption stage (Figure 4), two reinforcing loops were identified. The loop supports the circular economy concept. Postconsumption infrastructure supports the implementation of the EPR scheme, and mandatory EPR can expedite the setting of waste infrastructure. Both EPR and waste infrastructure enable DRS, which is a key enabling factor for a zero-waste business model. The zero-waste model can, in turn, expand the implementation of the DRS, as well as reduce waste in the environment. Household waste sorting rate is reinforced by green awareness, marketing and communication, regulation, and waste infrastructure. Household waste sorting rate and R&D of packaging can improve the SUP recycling rate and thus enable more waste to enter the circular system, resulting in less waste unmanaged. The zero-waste business model and the DRS can improve the brand image of the business. It can be observed that circularity can be achieved through the provision of waste systems and infrastructure, laws and regulation, R&D, and awareness raising.

### 4.2.5. QSDM Configuration

After the problems, actors, variables, system behaviors, and causal loops were comprehended, the QSDM was then revised and iterated through stakeholder interviews. CLDs and QSDM visualize how the systems interact. While QSDM can tell the direction of influences that occur in the system, behavior-over-time analysis allows us to depict the degree of such influences as well as the trend of the systems. According to the situational factors that instigated a change in consumption behavior, stock variables were expected to reach their peak during COVID-19, and shrink afterward. However, the stock levels after the COVID-19 situation are expected to be higher than the baseline, which was before

COVID-19. When the CLDs were integrated into QSDM (Figure 5), it was observable that the target initiatives influenced stock variables directly and indirectly. Significant mediator variables included cost, availability and quality of green packaging, consumer packaging demand, and a zero-waste business model. Government regulations and measures were assigned a more important role in steering the system. The importance of subsidies and incentives was stressed. Coordination among actors turned out to be one of the most important underlying factors in the system. Furthermore, factors that exhibited relationships across causal loops included the brand image of restaurants and food delivery platforms, green marketing and communication, and R&D on green packaging. Green marketing and communication influence the mental model, which greatly assists demand-driven sustainability. In terms of marketing implications, the brand image of restaurants and food delivery platforms can be influenced, positively or negatively, through their green actions. Likewise, R&D can improve the cost and quality of green packaging at the production stage, function and usage at the use stage, and recyclability at the postconsumption stage. Leverage points were extracted through the comprehension of this complex relationship between variables.

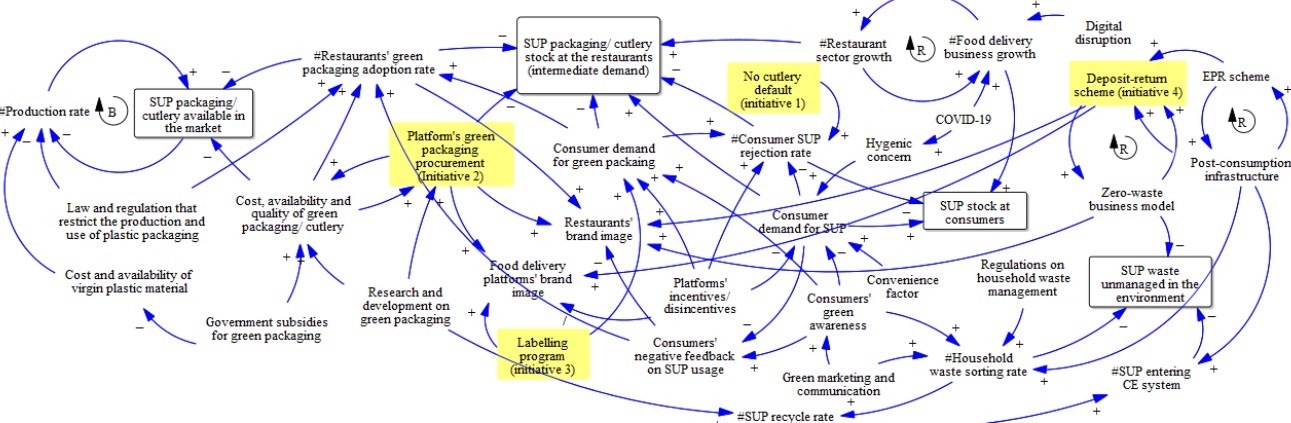

**Figure 5.** Qualitative system dynamics model of SUPs generated from Thailand's food delivery sector.

## 5. Discussion

### 5.1. The Leverage Points

Leverage points were derived from the results of thematic analysis and QSDM. The leverage points were analyzed from themes with high scores, as well as factors and variables that exhibited extensive linkages in QSDM, which could tackle problems identified in the system. As the diagrams and model were iterated, the leverage points became more apparent. A comprehensive analysis revealed that the leverage points included benefit alignment, cost and profit, postconsumption waste management systems, R&D, laws and regulations, and partnerships. The stakeholders agreed that improvement in these factors could significantly drive the system closer to the mutual goal.

#### 5.1.1. Benefit Alignment

Although benefit alignment is not explicitly presented in the model, it is the principle of the system that every party should hold to reach system self-organization. Incentive alignment was also highlighted under thematic analysis. Benefit alignment presumes that, if every party is satisfied with the benefits received, sustainable initiatives can be carried on without intervention. In systems thinking, Meadows and Wright [18] stresses that unaligned values may produce undesired system behavior. Benefits in the form of incentives act as behavioral shortcuts that can instigate changes in the system in which each actor makes preferable decisions. However, benefits need to be provided in the long term to sustain a sustainable system.

### 5.1.2. Cost and Profit

The results of both thematic analysis and QSDM reveal that cost and profit are the keys to a greener food delivery system. The key is to make alternative packaging cheaper or to make a profit out of the green initiatives. At the micro level, for-profit restaurants look for cheaper packaging options. Regardless of the presence of green intention, cheaper packaging of any kind is preferable, given that the quality is acceptable. Commercially, business ideas that are viable and have high return potential will gain acceptance and will drive the system towards a greener economy. Economic instruments such as subsidies and tax incentives can lower the cost of green packaging and influence production and consumption decisions.

### 5.1.3. Postconsumption Waste Management System

Because a consumption reduction strategy cannot offer overall benefit to the system, the analysis points toward a postconsumption waste management system as the key to circularity. Waste management infrastructure at the local level needs to be efficient and accessible. Mandatory responsibility at the household level needs to be clearly announced. If every household is required to be responsible for their own waste, the consumption decisions will be more conscious. Waste facilities need to be developed in parallel with behavioral change. Specifically, the waste management infrastructure should be ready to support the sorted waste. The allocation of facilities and infrastructure should be efficient enough to maximize waste entry into the circular economy.

### 5.1.4. Research and Development (R&D)

Although R&D did not dominate stakeholder's conversation in the thematic analysis, it exhibited significant influence across stages in QSDM. R&D on green packaging can increase its quality and reduce its cost. R&D on materials and production can offer new packaging solutions and increase the practicality of alternative packaging. It can improve recyclability and reduce the possibility of SUPs entering the recycling system. For example, mono-material packaging will improve its recyclability rate. Liu et al. [11] agreed that R&D on packaging material is one of the intervention points that can create sustainable food delivery ecosystem.

### 5.1.5. Laws, Regulations, and Standards

Regulation and standards of packaging production, use, and disposal were highlighted in both thematic and QSDM analysis. Laws and regulation control what should be available in the market at what price and for what amount. The measures include regulation on food packaging production such as deregulation on recycled material in food containers. The government can also regulate household waste management as well as EPR implementation. Furthermore, the official authorities should regulate the use of eco-labels on packaging. Standards on food packaging materials and environmental attributes such as biodegradability should be revised. Laws and regulation on producer responsibility such as EPR, including DRS, should be enacted. However, regulation should not excessively increase the financial burden or constrain the managerial decisions of private actors. The cost of compliance may discourage companies with limited financial resources from investing in sustainable practices. In Thailand, existing laws and regulation on plastic waste face implementation challenges, especially when it comes to the food packaging that involves small street vendors. EPR for plastic waste can only be implemented voluntarily. The ban has no legal effect. Tax incentives are not attractive. Therefore, while laws and regulation are being developed, market-based instruments should be applied in parallel.

### 5.1.6. Partnerships and Cooperation

Partnerships and cooperation are core to multi-stakeholder systems. Partnerships can leverage the system at various stages, and every SUP reduction initiative requires partnership and cooperation. Change at the system level takes time and requires extensive

actor engagement. For instance, partnership has been proven to be a key success factor of the deposit return model in many other countries. Likewise, packaging procurement initiatives were made possible through strong partnerships between packaging producers and food delivery platforms. The details of how partnerships can support each strategy are discussed in the next section.

*5.2. System Intervention Strategies*

The leverage points were interpreted into system intervention strategies. As shown in Figure 6, the strategies were classified as market-based instruments, system and infrastructure provision, and behavioral instruments. Each strategy was assigned potential and a time scale to implement. The potential was based on the initiative evaluation results, while the time scale was proposed according to the thematic. The analysis suggests that some strategies might have a higher potential to be implemented ahead of others. In the initial stage, nudge, labelling, and subsidies by private sectors should be implemented. In the next phase, the government should intervene in the system through price correction strategies and the provision of efficient waste infrastructure. R&D, laws, regulation and standards, and partnerships and cooperation act as the measures that could support every strategy at every stage. (R), (L), and (P) were assigned to each strategy to represent possible supportive measures. The legend on the left shows key actors who should be responsible for the implementation of each strategy.

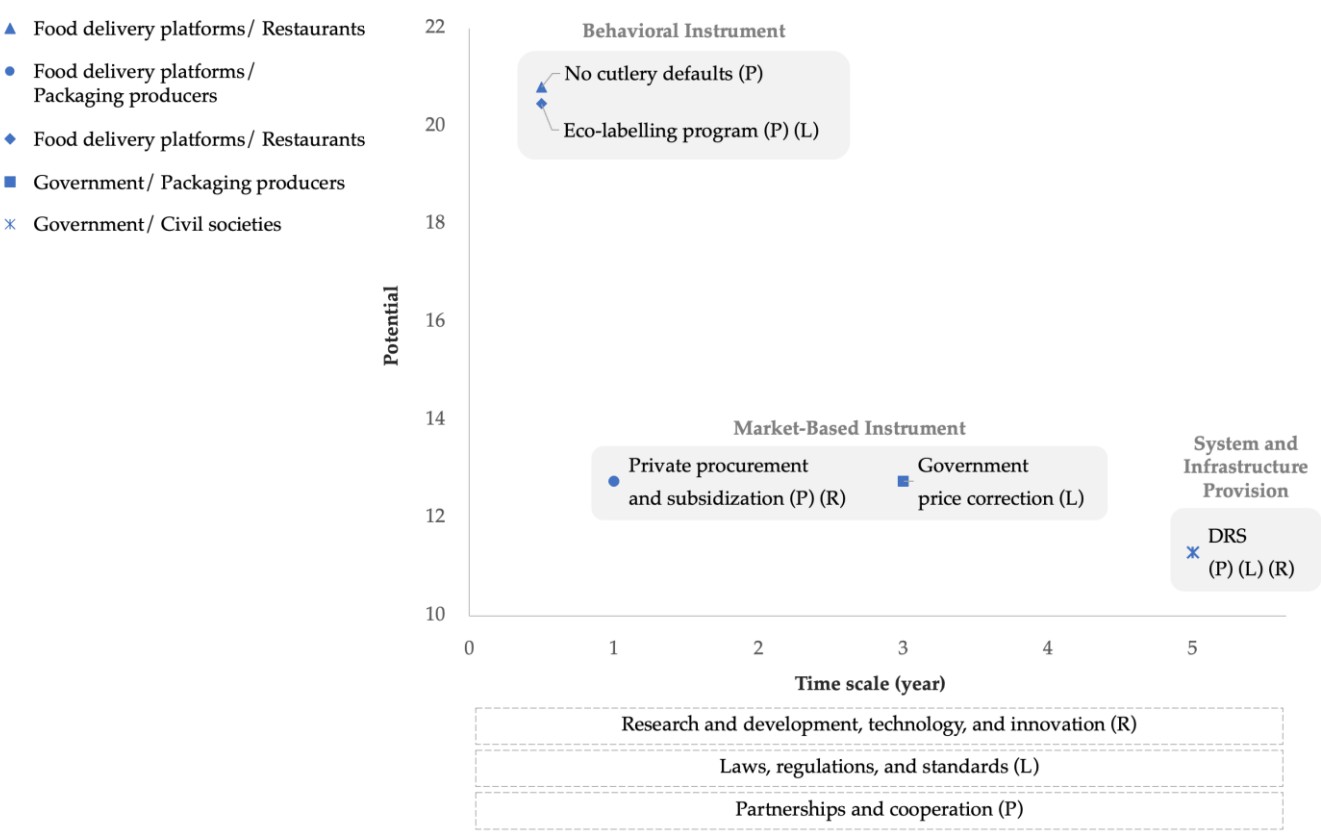

**Figure 6.** System intervention strategy framework.

5.2.1. Mid-Term Strategies

1. Behavioral Instruments: No cutlery default and Eco-labelling program

Nudge acts as a behavioral shortcut, which leads to behavioral change. In food delivery mobile applications, the 'no cutlery' option should be set as default. Considering Thai gastronomy, this function can be extended to cover condiment sachets to help reduce small and excessive packages. Cooperation and communication among food delivery

platforms, restaurants and consumers should be well-established. Restaurants should be careful with their cutlery give-out routine. In addition, eco-labelling acts as an information instrument which can influence consumption decisions. A green restaurant labelling scheme is expected to improve consumer decisions. Food delivery platforms should consider giving incentives and rewards to restaurants and consumers who participate in the program. The government should regulate and set standards for green restaurants and eco-labels. Furthermore, green marketing and communication could promote awareness, values, and attitudes, which lead to behavioral change. Both measures were found to have high potential and can be implemented promptly. They have a high chance of success, high expected outcome, and require fewer resources.

2.  Market-based Instruments: Private procurement and subsidization

Although the packaging producers, delivery providers, restaurants, and consumers are expected to evenly absorb extra costs to internalize the externalities from their practices, the analysis highlights the unbalanced capability and interests of each actor, leading to unaligned benefits. Mid-term strategies, therefore, propose that food delivery platforms procure and partially subsidize green packaging to help their restaurant partners with limited financial capability. Partnerships between food delivery platforms and packaging producers can facilitate the scheme, which can yield marketing benefits to the firms. Furthermore, R&D can result in new alternative packaging solutions, which would increase their presence in the market. The program can be implemented in a year. However, it has relatively low potential as it relies solely on private budgets. Government subsidies are expected to yield a higher contribution to the system. Alternatively, the government can support partnerships among private actors to make packaging procurement schemes sustainable enough to influence the systemic structure.

### 5.2.2. Long-Term Strategies

1.  Market-based Instruments: Price correction

Market-based initiatives carried out by private actors only led to limited, short-term results. Therefore, the government is expected to be a key actor to improve market failures. Pricing mechanisms can lower the cost of preferable alternatives. Meadows and Wright, and Seyfang [18,52] stress the importance of 'price signals' as a leverage point that keeps the balance of demand and supply in the market system. Currently, a 25% corporate tax exemption covers 11 types of compostable plastic products, which include food packaging, straws, and cutlery. However, the analysis revealed that this scheme faces government red tape. The tax exemption level is also not compelling. Therefore, the government should consider increasing the exemption level and expanding the list of eligible alternatives, as well as improving the submission process. The current tax exemption measure ends in 2024. Another stage that could benefit from intervention would be the taxation of unnecessary SUP packaging as a disincentive for the producers, restaurants, and customers to avoid SUPs. Regulatory measures take time to enact. The government could take three years to review the current tax exemption scheme, as well as introduce a new tax penalty scheme. These measures would enable the price signal to be precisely communicated to both ends. Likewise, it supports the cost minimization and profit maximization of private companies. Furthermore, through partnership, benefit alignment could be expected.

2.  Infrastructure and Systems Provision

The provision of postconsumption waste management systems is a key leverage point in the system that would improve the rate of proper plastic waste treatment. Proper and adequate waste facilities at household level as well as at the disposal stage would enable the system's circularity. However, infrastructure provision needs to be coupled with laws and regulation, especially on EPR schemes, to ensure efficient outcomes. However, DRS is particularly challenging in a convenience-based business ecosystem such as food delivery service. This initiative requires resources, as well as action and cooperation from extensive stakeholders, especially the food delivery platforms, civil society actors, and consumers.

The analysis reveals that, if successful, DRS would create an impactful outcome. Thus, through partnership, the food delivery platform, as the logistic provider, could initiate a pilot DRS project in their service areas. The government could provide drop sites. Civil society actors can coordinate and promote the scheme. R&D on packaging can improve its durability, application, and recyclability. Additionally, R&D on waste logistics and consumer behaviors could be supportive.

## 6. Conclusions

It can be observed that the leverage points found in this study are points that were recurrently mentioned at systemic structure level according to the systems thinking framework. Likewise, in system dynamics analysis, the leverage points exhibited several linkages with other variables, representing their power to influence the systems. To conclude, voluntary programs can only be sustained if each party receives satisfactory benefits from the change. Cost and profit are the fundamental concerns of for-profit organizations. If prices are corrected through market-based instruments, the greener choices will be those that are cheaper. The interventions allow the market system to adjust itself toward the optimum point where prices reflect the true cost of the product. In that case, the private sector will find sustainable practices profitable and carry on such practices. Furthermore, the improvement of the postconsumption waste management system is a key leverage point toward the circular economy in Thailand. EPR should be legally enforced, and the deposit return model should be widely promoted. While these policies are being developed, private entities could voluntarily apply behavioral instruments together with marketing incentives to attract restaurant partners and consumers to opt for greener choices. Moreover, partnerships are the key to systems' accordance and alignment. New packaging solutions can be achieved through R&D. Laws, regulation, and standards should govern how packaging is produced, consumed, and discarded.

This research provides a theoretical contribution to the understanding of complex systems where economic benefits and sustainability find their balance. Overall, it contributes to Thailand's efforts in reducing food delivery SUP waste. The case study contributes to the green growth model where the demand for corporate responsibility arises. It also provides useful insights for policy makers of other developing countries to develop measures that tackle SUP emerging from this fast-growing business in a post-COVID-19 future.

**Supplementary Materials:** The following supporting information can be downloaded at: https://www.mdpi.com/article/10.3390/su14159173/s1, Table S1: Behavior-over-time calculation table; Table S2: Interview with stakeholders and questions.

**Author Contributions:** Conceptualization, B.W. and S.V.; methodology, B.W. and S.V.; formal analysis, B.W.; investigation, B.W.; supervision, S.V.; writing—original draft preparation, B.W. All authors have read and agreed to the published version of the manuscript.

**Funding:** This research was funded by The 100th Anniversary Chulalongkorn University Fund for Doctoral Scholarship (GCUGE11) and Graduate School Thesis Grant (GCUGR1225643018D), Chulalongkorn University.

**Institutional Review Board Statement:** Not applicable.

**Informed Consent Statement:** Not applicable.

**Data Availability Statement:** Not applicable.

**Conflicts of Interest:** The authors declare no conflict of interest.

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
