# Peer review of "A Systems Thinking Approach towards Single-Use Plastics Reduction in Food Delivery Business in Thailand"

_sustainability, doi:10.3390/su14159173_

Round 1

Reviewer 1 Report

This is an interesting study. I am not familiar with the methodology of system dynamics but I see value in the exercise.  Ultimately the key leverage points which emanate from the analysis are the obvious ones but this brings added credibility and strength to the insights and recommendations.  The paper is extremely well written and insightful.  

Author Response

This is an interesting study. I am not familiar with the methodology of system dynamics but I see value in the exercise.  Ultimately the key leverage points which emanate from the analysis are the obvious ones but this brings added credibility and strength to the insights and recommendations.  The paper is extremely well written and insightful.  

Response: Thank you for your comments. System dynamics under a system thinking approach can deliver an insightful understanding of how the system operates and how change at one point can affect others. The leverage points may be obvious but are often neglected when being viewed from a compartmental perspective.

Reviewer 2 Report

This study attempts to identify important leverage points for reducing the use and waste of SUPs, which are of concern globally these days. System thinking approach is important, so I can understand why the authors conducted this study. However, there are several drawbacks in the current version of the manuscript: lack of clear explanation about the procedure of this study or the structure of this manuscript, insufficient justification (verification) of the results of QSDM, an issue about discussion based on the results and complexity of the system, etc. I have to say that this version requires a major revision.

Specific points are as follows:

1)    Abstract

(a) Key leverage points: The five LPs identified are too general. Experts in this field understand their importance. So, what are the true contributions of this study? The main results do not seem that these conclusions are not so based on the authors’ QSDM.

More specific results are useful for the experts. For example, this study addresses the four target initiatives. Hence, concrete findings about these initiatives are based more on the authors’ QSDM and the authors’ contributions are clearer. (See the last comment also)

(b) The sentence starting with “In addition” (The 5th line from the bottom.): This study does not stimulates policy initiatives (At least, yet. This study is not even published. How can it affect initiatives? Therefore, this part should be removed, i.e., “In addition, this study suggests that system intervention strategies ....” Or the sentence should be “In addition, to stimulate policy initiatives this study suggests that system intervention strategies ....”

2)    Lack of clear explanation about the procedure of this study or the structure of this manuscript.

In my understanding, this study conducted literature review and stakeholder interviews, and then based on these results QSDM and thematic analysis were done. If so and this is appropriate, Section 3 should come before the current Section 2 and explain the procedure of this study: literature review, stakeholder interview, QSDM, thematic analysis, and discussion. It is a good idea to show a flowchart of this procedure. Otherwise, the authors should explain the structure of sections in this manuscript at the end of Section 1. Currently, it is not clear how the results of the literature review are linked with the subsequent sections.

3)    Section 2.1: Inappropriate reference

Social practices theory is an important theory in the filed of sustainable consumption and it also pays attention to systemic structure. However, the results of QSDM do not reflect phenomena of social practices. That is, the results of QSDM does not include causal relationships about how everyday practices in turn influence human behaviors, nor how the stock variables influence stakeholders’ behaviors (Figs 2-5).

Hence, Section 2.1 should be removed and social practice theory should be referred to in Introduction section briefly.

If the authors insist to keep Section 2.1 in the current manner, I demand that the results must indicate phenomena of social practices.

4)    Section 2.3: Lack of explanation about the benefits and contribution of a qualitative approach.

(a) I understand that existing similar studies are based on a quantitative approach by reading this section. However, the authors fail to explain what are the benefits of a qualitative approach. Without it, I do not understand why the authors had to apply a qualitative approach in this study. A clear explanation is necessary. (Is it a wide coverage of the causal relationships?)

(b) This question brings another point. The results of QSDM fits to the motive of applying a qualitative approach. Which parts of Figs 2-5 were not studied by the previous studies?

5)    Headings in Section 2.4

The headings are not so accurate and understandable for readers. In my understanding, Section 2.4 is to determine the target initiatives of this study, and for the purpose, the authors reviewed relevant policies and actions in Sections 2.4.1 and 2.4.2.

If so, the heading of Section 2.4 should be “Initiatives in Food Sector for sustainability” and that of Sections 2.4.1 should include “initiatives” and 2.4.2 should be “Private-led initiatives.”

(“Food delivery sector” refers only to delivery from a shop to end consumers. (“Food provision” has a wider meaning.) As it includes manufactures and retailers, it should be “Food sector.” “Sustainable effort” is too vague.)

6)    Line 7, Section 2.4.1

(a) I opposed to “most measures were formed based on the Extended Producer Responsibility (EPR) concept”. Ban is not based on EPR, which is regulatory policy instruments. Economic policy instruments and awareness raising of consumers are used for SUP as the authors themselves explain the current policy situation. Rewrite this sentence.

(b) In addition, EPR often refers to manufactures only and some readers may think that retailers are not included. The authors’ definition and understanding of EPR seem to be including retailers. Please explain that the authors use EPR in a broader meaning.

7)    Section 3: Unclear explanation about the linkage between interviews and QSDM and thematic analysis

Readers as well as I do not understand how QSDM and thematic analysis were done based on the results of the interviews. Please explain it more clearly so that readers can conduct a similar study; in other words, the current explanation lacks sufficient level of transparency and traceability of research. For example, did the authors made Figs 2-5 only from what stakeholders mentioned? How did they check the accuracy of what they said? Were there any additions by the authors? Were there any contradictory remarks between the respondents? I understand that it is difficult to generalize these aspects and procedure but more explanation (to some possible extent) is necessary.

8)    Table S2

Column heads must be provided for each column.

9)    Sections 3.2 and 4.1

(a) The authors used the term “thematic analysis” but used the term “system thinking analysis”. Are they the same? What are the differences? I am confused with the different terms.

(b) I do not follow what the authors mean by “thematic analysis.” I might be unfamiliar with it. Isn’t it just coding of what the respondents said? “Coding of interview results” is much more understandable for many readers.

(c) Table 1 is totally unclear to me because it fails to deliver what are the meaning of each column and the bold headings. The heading of Table 1 is too rough, in addition. Explain what is what in the table. Especially the left column, the middle column, and right column. For me “Theme” is not accurate nor understandable. Shouldn’t it be “item” or “category”?  

(d) How many respondents tell something about each themes?

(e) Table S3: What is the meaning of bold words? Are they coded ones? The overarching column head is missing, which should be “stakeholder.” The column head the far-left column is missing, too. Is it “Theme”, right?

10) Section 4.2.1: The first sentence “The first step of constructing QSDM is to identify problems in the system.”

How many problems were identified? What are they? Are the problems included in Figs 2-5?

11) Section 4.2.2

This section is not the results. It is about the method and procedure of how the authors analyzed the interview results to show Table 1 and Figs 2-5. Move to Section 3.

12) Section 4.2.3

(a) I think this result is irrelevant to the main results of this study. It does not bring a meaning conclusion. Just SUP waste is increasing. Moreover, no equation is provided. Inappropriate for a scientific paper. Furthermore, no mention in the method section about this calculation and how this calculation is linked with the other results (See my comment 4). I suggest deleting this section.

(b) The fourth line from the bottom, “In conclusion, ....”: This is not the result. The author assumed that SUP waste generates according to demand and then calculated it. Figure 1 does not say anything about demand or supply is a main driver. This sentence should be deleted.

13) Figs. 2-5

May arbitrariness are found in these figures. Elaboration is necessary.

(a) Shading of initiatives is halfway: For example, “Law and regulation ...” on the left of Fig. 2 and “Green marketing ...“ at the bottom of Fig. 3 are interventions. All of the intervention and initiatives should be shaded.

(b) Unclear mass balance of stock and flows: Material flow and mass balance are missing in these figures. Does any problem occur without it?

(c) Validation of the results: in relation to the comment (b), how did the author validate the accuracy of the results? For example, I have a different opinion about the reinforcing loop in Fig. 3. The amount of meals people is constant basically. If food delivery (buying and earing at home) increases, then eating out will decrease. Why do these two increase endlessly? “COVID” reduced the frequency of eating out of people. Why isn’t there an arrow from “COVID” to “Restaurant sector growth”? “Consumer demand for green packaging” is a behavior of green consumer. Don’t they reduce the use of SUP? That is, an arrow from “Consumer demand for green packaging” to “Consumer SUP reduction rate” should be added.

(d) Distinction of direct influence and indirect influence: For example, in Fig. 4, “household sorting” influences “SUP entering CE system” through “SUP recycling rate”, not directly. So, the arrow from sorting to CE system should be removed. In addition, how did the authors distinguish direct and indirect influences when drawing CLDs (Causal Loop Diagrams)?

(e) Unclear distinction of stock and flows: I believe that the authors fail to distinguish the level and the capacity (maximum value) of stock variables. The stock level is influenced by flow variables only. The capacity of a stock variable is influenced by factors. However, it is unclear that arrows to stock variables are affecting to either of them in the current version of the figures. And some factor influence to both flow and stock (e.g., “Law and regulation ...” influences a flow, production rate, and stock, “SUP packaging available in the market”. Confirm the arrows are written without confusion.) Clearly explain this distinction and make figures in such a way without inconsistency. I recommend to distinguish flow variables and factor variables.

(f) The expression of two stock variables seem as if they are flow variables. Shouldn’t “SUP distributed to consumers” and “final SUP waste unmanaged” be “SUP stock at consumers” and “SUP waste unmanaged in the environment”, respectively?

(g)end must be provided.

(h) Signs of some arrows are missing or cannot be seen. Make sure all arrow has a sign.

(i) Some important factors are missing in a figure although they are included in another figure. I have to say that “Green marketing” in Fig 3 is a strong driver for “household sorting” in Fig 4. I think that these are included in Fig 5; however, to understand behaviors in a different stage, these should not be removed from Figs. 2-4.

14) Fig. 5

This is the complex mechanism what the authors advocate to understand. However, discussion about complex mechanisms and the loops in this complex diagram are not presented. Section 4.2.5 should be added more discussion.

15) Section 5.1

It is unclear how the authors identified LPs. What are the LPs in Figs 2-5? Sections 5.1.1 and 5.1.2 are about how to find important LPs, which should be explained in the method section. Sections 5.1.3 and 5.1.4 are too general, which is seemingly not so based on the CLDs. More explanation and discussion based on the authors’ original results of Figs 2-5 are needed.

16) Section 5.2

This section is too general for me. I think that this section can be written based on the literature review without looking at CLDs. If so, this is contradictory for the authors’ intent: more comprehensive understanding of the complex mechanisms is necessary. In light of the motive and objective of this study, discussion should be more oriented from the complexity of the system. (Sorry if I am wrong. At least, the current version of the manuscript has a weak linkage between CLDs and discussion in Section 5.2.

Reviewer 3 Report

The article is scientifically modest and concerns theoretical considerations on the methods of reducing plastic package from food. This issue is already widely know.

The authors of the article based their considerations mainly on reports (marked in blue).

There are no questionnaire research in the article that would show the trends and habits of the people of Thailand in in the field of food purchases.

The authors should enrich the article with questionnaire research. The theoretical part should be based mainly on articles in international journals.

Author Response

The article is scientifically modest and concerns theoretical considerations on the methods of reducing plastic package from food. This issue is already widely know.

The authors of the article based their considerations mainly on reports (marked in blue).

There are no questionnaire research in the article that would show the trends and habits of the people of Thailand in in the field of food purchases.

The authors should enrich the article with questionnaire research. The theoretical part should be based mainly on articles in international journals.

Thank you for your comments. 

Response 1: The system thinking approach utilises policy information at every level. The reports from different organisations need to be reviewed in order to understand the holistic interconnections in the system. Additionally, there are a limited number of studies on SUPs from food delivery services conducted at a system level. The majority of them explore pro-environmental behaviours or the commercialization of green packaging. Therefore, this study mostly relies on government, academic, and business reports. However, among the marked references, only reference numbers 1-10, 12-13, 44, and 52-57 are report papers. Reference numbers 17-18, 20-21, 26-27, 30, 32-33, 45-46, and 49-50 are published articles, books, conference papers, and working papers. The theoretical part (2.1-2.3) contains reference numbers 13-41, reference numbers 14-16, 19-20, 23-25, 28-31, 33-37, and 41 (18 out of 29) are articles in international journals. As some parts of section 2.1. “Social Practices and the Systems of Provision Theory” and section 4.2.3 “Behavior-Over-Time Analysis” were removed (according to reviewer 2’s comments), some references were removed from the list. Minor details have been added to the references where applicable, e.g. volume, issues and page number. (Please note that the section numbers, reference numbers, and reference orders have changed in the resubmitted version. Please refer to the first version).

Response 2: This manuscript is part of a doctoral dissertation, which is divided into two parts: consumer research and systems research. The first part explores consumers’ behavioural, psychological, and demographic profiles and segments them accordingly. In this section, online and paper-and-pencil surveys were conducted to observe consumers’ food delivery usage behaviour before, during, and after the COVID-19 lockdown measure. The results revealed that the food consumption behaviour of Thai consumers shifted towards online delivery. Marketing and policy implications were delivered. However, while the first study focuses on the demand for consumption, the current manuscript aims to understand the dynamics and structure of the system, which involves many actors. Therefore, in-depth interviews were conducted.

Round 2

Reviewer 3 Report

Thank you for completing the article. I accept the article in present form.